# Trends in the levels, causes, and risk factors of maternal mortality in Pakistan: A comparative analysis of national surveys of 2007 and 2019

**Farid Midhet[1,2], Samina Naeem Khalid**  **[2]\*, Shehla Baqai[1], Shahzad Ali Khan[2]**

**1** Public Health Department, Bahria University Karachi Campus, Karachi, Karachi City, Sindh, Pakistan,
**2** Public Health Department (MNCH), Health Services Academy, Islamabad, Islamabad Capital Territory, Pakistan

\* drsamina@hsa.edu.pk

## Abstract

### Background

Maternal mortality ratio (MMR) has decreased worldwide but Pakistan is still striving towards achieving the SDG targets for maternal health. This study highlights the trends in maternal mortality levels and risk factors in Pakistan between 2007 and 2019.

### Methods

This study compares the results of secondary data analysis of the Pakistan Maternal Mortality Survey 2019 with the Pakistan Demographic and Health Survey 2007. A nested case-control study was carved to compare maternal deaths with the women who survived a pregnancy, in the same sampling clusters during the same period. Logistic regression was used to estimate odds ratios (OR) for major risk factors of maternal mortality after adjusting for the women's age, parity, education, and wealth quintile.

### Results

In 2019, Pakistan's MMR was 186 per 100,000 live births, registering a 33% decline from 2007 (rural 42% vs. urban 11%). The leading causes of maternal mortality were postpartum hemorrhage, hypertensive disease of pregnancy, postpartum infection, and post-abortion complications. Women > 35 years and those expecting their first child were more likely to die from childbirth, while those who had ever used family planning had a lower risk according to the data for both years. In 2007, a distance of > 40 kilometers to a hospital significantly increased the risk of mortality but this association was not significant in 2019. In 2019, women who died were more likely to receive antenatal care than those who survived (adjusted OR 9.3); this association was not significant in 2007.

### Conclusion

The modest reduction in MMR can be attributed to improved access to maternal health services in rural areas with increased antenatal care and institutional deliveries. However, most

**Data Availability Statement:** https://nips.org.pk www.DHSprogram.com.

**Funding:** The author(s) received no specific funding for this work.

**Competing interests:** The authors have no competing interests.

maternal deaths were caused by poor accessibility to quality emergency obstetric care. Lack of family planning remains a major risk factor for high maternal mortality in Pakistan.

## Introduction

The Sustainable Development Goals (SDGs) expanded upon the international pledges in 2015 that the countries made to the Millennium Development Goals (MDGs) in 2000. Goal 3: "Ensure healthy lives and promote well-being for all at all ages" is one of the three health-related SDGs [1–3]. The World Health Organization (WHO) established a consensus statement defining the approaches to end preventable maternal mortality (EPMM) at about the same time. The SDG target 3.1, which was adopted by EPMM, pledges to lower the global maternal mortality ratio [MMR] to less than 70 deaths per 100,000 live births by 2030 [4].

Achieving the SDGs requires significant global efforts to strengthen the healthcare system and ensure that everyone has access to quality healthcare [5]. The MMR is an important factor in the Human Development Index (HDI), a metric for assessing a nation's economic development [6]. The MMR is calculated as the division of the number of maternal deaths during a given period of time by the number of live births over that same period and expressed as maternal deaths per 100,000 live births [7].

Maternal death is defined by the International Classification of Diseases, 10th version (ICD-10) as "the death of a woman while pregnant, or within 42 days of termination of pregnancy, from any cause related to or aggravated by the pregnancy or its management but not from accidental or incidental causes" [8]. This is regardless of the geographic location of maternal death or the current circumstances of pregnant women. On the other hand, maternal health includes the well-being of women during all stages of pregnancy, childbirth, and the postpartum period [9]. Maternal and perinatal health, according to WHO, are among the most crucial issues to affect the world's economic burden [10].

While the MMR has decreased globally, it is unacceptably high in many developing countries including Pakistan, where high MMR continues to be a major source of public health concern [11]. Over 99% of maternal deaths occur in low- and middle-income countries, particularly in their impoverished communities and remote rural areas [12]. Hemorrhage, sepsis, difficult births, and hypertensive disorders are the most prominent causes of maternal mortality [13]. One in three maternal deaths is hemorrhage-related, which is a warning indication because postpartum hemorrhage (PPH) is much more common in poor countries [14].

The primary reason behind the recent global decline in MMR is improved access to family planning, skilled birth assistance, and emergency obstetric and newborn care (EmONC) [15]. The remarkable improvements in communications and transportation across the globe over the last decade may also have been responsible for the decline in MMR: UN inter-agency statistics show that between 2000 and 2017, the global MMR declined by 38%, from 342 to 211 maternal deaths per 100,000 live births–an average annual decrease of 2.9% (WHO 2019) [9, 16].

Pakistan is lagging behind other countries in achieving its SDG targets in maternal health. According to the most recent data, the MMR in Pakistan was 186 maternal deaths per 100,000 live births in the period 2017–2019, registering a 33% decline from 276 in 2005–2007 [17, 18]. A majority of reproductive-age women in Pakistan are uneducated, undernourished, and anemic [19]. They also give birth to a high percentage of preterm and low birth weight babies in settings with insufficient and inadequate maternal and newborn care. Hence, the rates of

perinatal and neonatal mortality in Pakistan are also higher than in its neighboring countries. The situation of maternal and newborn health indicates that Pakistan has a long way to go in improving the quality and accessibility of its maternal, newborn, and child health (MNCH) care [20].

The Pakistan Demographic and Health Survey from 2007 and the Pakistan Maternal Mortality Survey from 2019 are two nationwide surveys that were conducted 12 years apart [7, 18]. The results of their in-depth secondary data analysis on the risk factors of maternal mortality are compared in this study. To determine the areas that require intervention, we compare the levels, causes, and risk factors of maternal mortality between the two surveys.

## Materials and methods

Pakistan has conducted two nationwide surveys to determine the extent of and contributing factors to maternal mortality. The first was a part of the Pakistan Demographic and Health Survey (PDHS) in 2006–07 [21], and the second was the Pakistan Maternal Mortality Survey 2019 (PMMS), a national survey exclusively on maternal mortality and morbidity [7]. National Institute of Population Studies (NIPS) https://nips.org.pk, the sole public sector population research organization, carried out these surveys with technical assistance from the Demographic and Health Surveys Program of USAID. Both surveys estimated the MMR from nationally representative samples and collected detailed information on the medical causes and biological and socio-demographic risk factors of maternal mortality.

We compare the rates, causes, and risk factors of maternal mortality between 2007 and 2019 to highlight the trends that could be helpful for planning and program development. In 2009, NIPS published a comprehensive report of in-depth analysis of 2007 PDHS data, which included a chapter on maternal mortality. Using the same methodology as that used for the in-depth analysis of PDHS 2007, we performed a secondary analysis of the PMMS 2019 data on maternal mortality, which is used in this paper for the trend analysis. The final reports of the Pakistan DHS 2007 and PMMS 2019 contain details about the sample and survey methods. These reports are accessible from the DHS Program website www.DHSprogram.com [7, 21, 22].

### Sampling methodology

A two-stage cluster sampling process was used to pick the 95,441 households for the PDHS 2007 sample, which was limited to the four provinces (Baluchistan, Sindh, Khyber Pakhtunkhwa-KP, and Punjab) and excluded the territories of Azad Jammu and Kashmir (AJK) and Gilgit-Baltistan (GB). For every household in the sample, information on births and deaths that occurred during the past three years was collected. For all female deaths between 15–49 years, interviews with the deceased woman's next of kin were completed by specially trained interviewers and using verbal autopsy (VA) questionnaires, to investigate the cause of death further.

Panels of medical experts, including OB/GYNs, specialists, and general practitioners, reviewed the completed VA questionnaires. They determined the immediate and underlying causes of death according to ICD-10 and classified them as direct/indirect maternal deaths and deaths due to non-maternal causes. In a 10% sub-sample (9,255 households with 10,023 ever-married women), detailed household and women's questionnaires (demographic and socioeconomic characteristics and pregnancy history) were administered using the standard methodology of demographic and health surveys.

The PMMS 2019 sample comprised 136,226 households in the four provinces as well as the territories of AJK and GB. For this analysis, however, only the data from the four provinces is included (108,746 households) to make it comparable to the in-depth analysis of PDHS 2007.

The process of identifying female deaths in PMMS 2019 was similar to that in PDHS 2007. There were 940 deaths of women in the 15–49 years age group in the four provinces, which were further investigated using the VA questionnaire, followed by reviews by panels of medical experts who assigned the cause of death according to ICD-10, as in PDHS 2007. Similarly, in a 10% sub-sample (10,479 households), 11,859 ever-married women aged 15–49 years were interviewed to record detailed information on their demographic and socioeconomic characteristics and pregnancy histories of the past three years.

### Statistical analysis

The sampling design of the PDHS 2007 and PMMS 2019 was such that the main samples (95,441 households and 108,746 households respectively) were used only to elicit information on births and deaths (and to identify the maternal deaths) occurring during the past three years. Detailed information on household and women's variables was collected from the sub-samples (10% each). Hence, it was not possible to directly estimate the risk of maternal mortality according to biological, demographic, and socioeconomic factors because this information was available only from the 10% sub-sample. To overcome this difficulty, an unmatched nested case-control study was designed for the in-depth analysis of PDHS 2007 data on maternal mortality, whereby maternal deaths were regarded as cases and the women surviving a pregnancy during the same period (three years before the survey) in the same sample clusters were regarded as controls. We adopted the same methodology for the in-depth analysis of PMMS 2019 data on maternal mortality: For PDHS 2007 data, maternal deaths (n = 230) served as cases while a random sample of the women who had a pregnancy in the last three years (n = 2,300) was selected to serve as controls. For the analysis of the PMMS 2019 data, we assigned all maternal deaths (n = 148) as cases and all the women who had a pregnancy in the last three years (n = 8,210) served as controls. Cases and controls were not matched. This sample size yielded a power of 88% to estimate the odds ratio of 1.75 or higher at a 95% confidence level.

Binary logistic regression (SPSS version 19.0) was used to estimate the adjusted odds ratios (AOR) for the association between the risk factors of interest and maternal mortality after controlling for the biological, demographic, and socioeconomic risk factors. Since the clusters were numerous (n = 1,096) and of very small size (mean number of cases and controls per cluster was 7.63), the intra-cluster correlations were close to zero and the need for generalized estimating equations (GEE) was not critical. Therefore, we used traditional binary logistic regression analysis for this analysis.

We compare the MMRs, the causes of death, and the biological and socioeconomic risk factors of maternal mortality from the two surveys. All the figures from PDHS 2007 reported here have been taken from the published reports referred earlier; while those from PMMS 2019 arise from the secondary analysis of raw data files, which were provided to us by NIPS and are also available on the DHS Program website.

### Results

The leading causes of maternal mortality were postpartum hemorrhage (PPH), hypertensive disease of pregnancy (HDP), postpartum infection, and post-abortion complications. In 2007, PPH caused 33% of maternal deaths, followed by infections (14%), and HDP (10%), while 5% of maternal deaths were due to complications of abortion. In 2019, PPH remained the leading cause of maternal deaths (41%), followed by HDP (29%); the proportion of abortion-related maternal deaths increased to 10% (Fig 1).

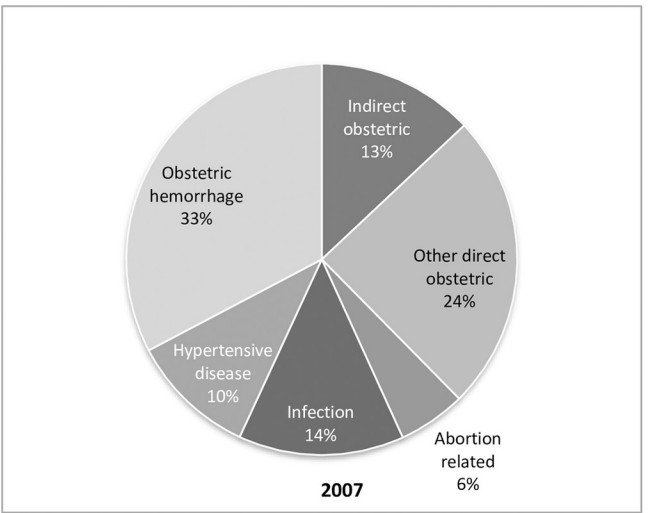
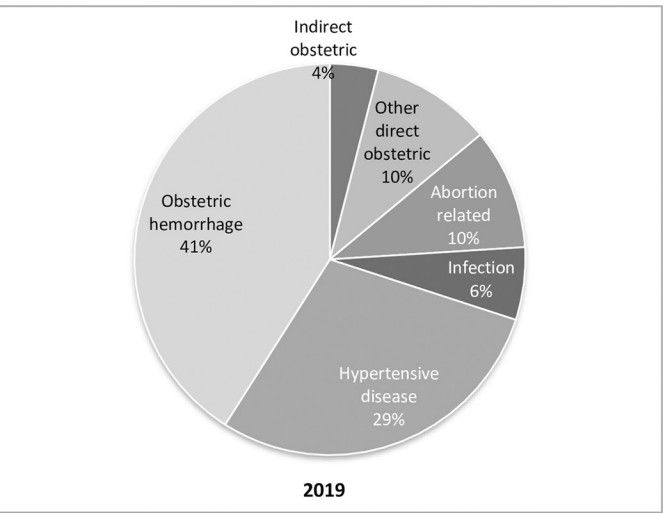

**Fig 1. Percentage distribution of maternal deaths by cause of death, PDHS 2007 and PMMS 2019.**

However, cause-specific MMR for PPH decreased from 90 to 76 (maternal deaths per 100,000 live births) and for infections from 38 to 11, while for HDP it increased from 29 to 54 (Fig 2).

Between 2007 and 2019, MMR in Pakistan declined by 33% from 276 to 186 maternal deaths per 100,000 live births (P = 0.049) (Fig 3).

The reduction in MMR was significantly greater in rural areas, eliminating the urban-rural difference, which was present in 2007 (320 in rural areas and 177 in urban areas) (P < 0.001). MMR in rural areas saw a 42% decline to 185 (P < 0.001), while MMR in urban areas declined by only 11% to 157 (P = 0.29). As a result, the gap in MMR between urban and rural areas disappeared in 2019 (Fig 4).

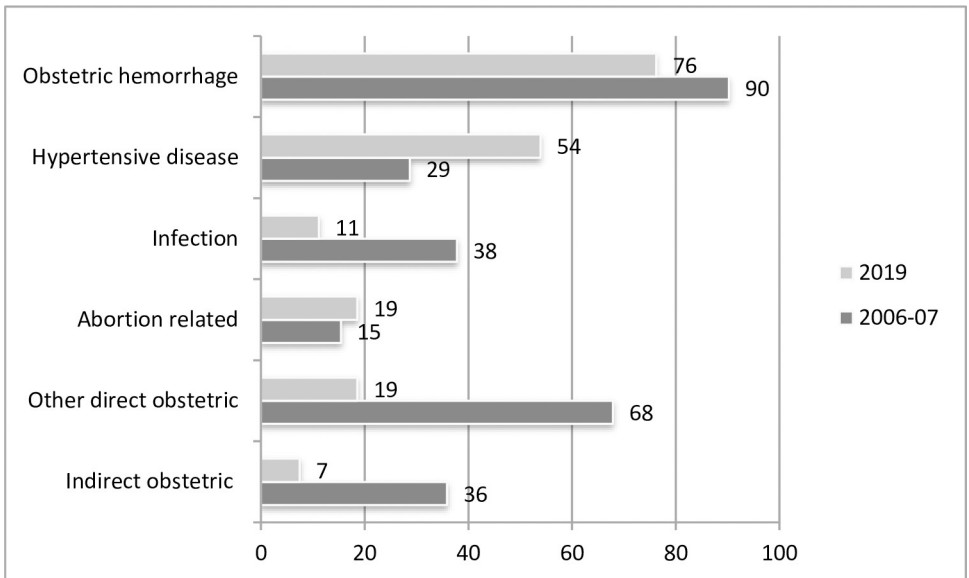

**Fig 2. Maternal mortality ratio (MMR) by cause of death, per 100,000 live births, PDHS 2007 and PMMS 2019.**

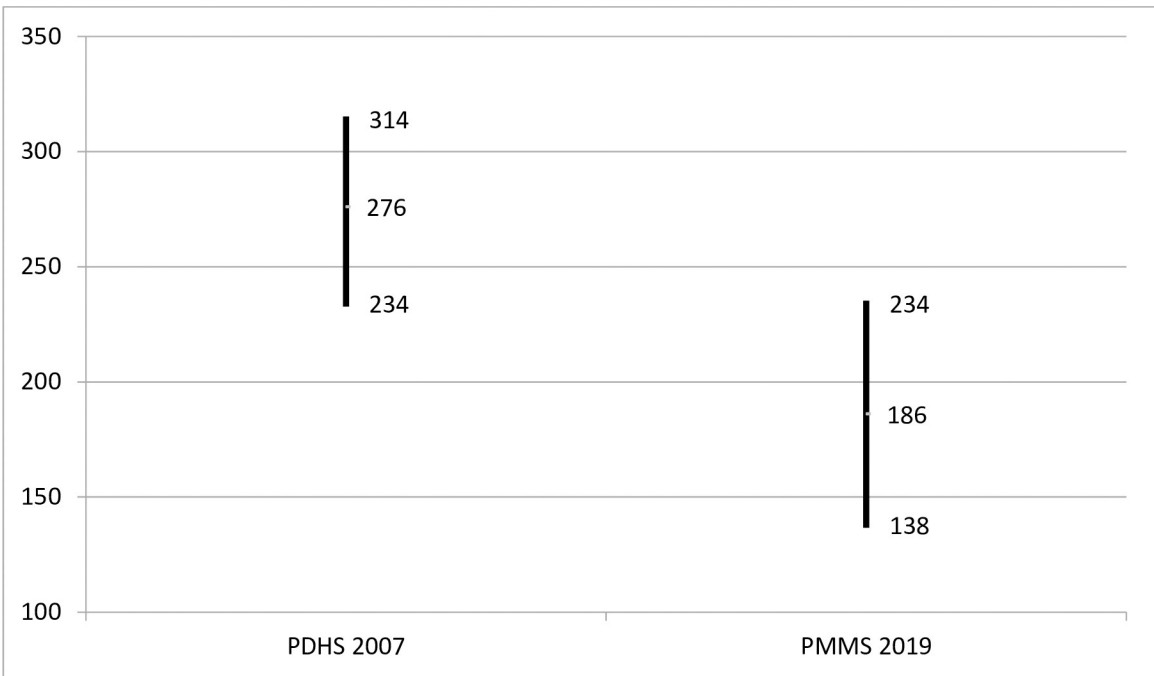

**Fig 3. MMR (maternal deaths per 100,000 live births) and 95% confidence limits, PDHS 2007 and PMMS 2019.**

A comparison of the risk factors of maternal mortality found in PDHS 2007 and PMMS 2019 helps explain the modest decline in MMR during this period. Tables 1–3 present the adjusted odds ratios (AORs) for selected biological, demographic, and socioeconomic risk

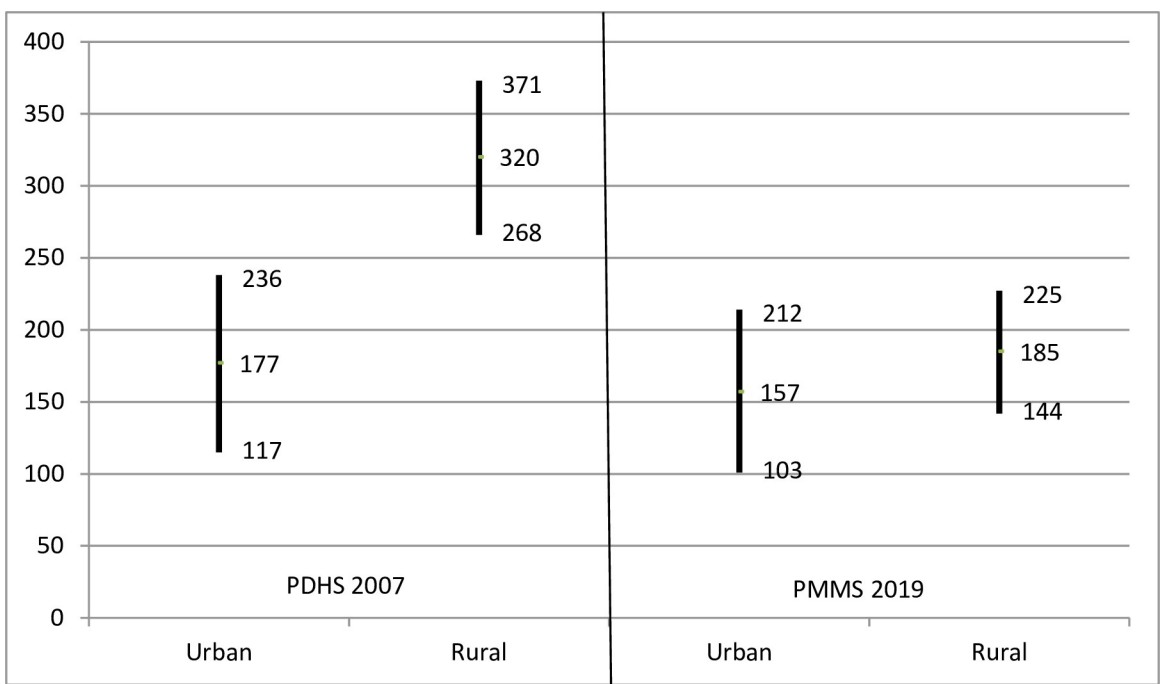

**Fig 4. MMR (maternal deaths per 100,000 live births) and 95% confidence limits by urban and rural areas, PDHS 2007 and PMMS 2019.**

**Table 1. Comparison of biological and socioeconomic risk factors of maternal mortality, in Pakistan (PDHS 2007 and PMMS 2019) (see note 1).**

| Model | Risk Factor | Adjusted Odds Ratio (95% CI) | |
|---|---|---|---|
| | | PDHS 2007* | PMMS 2019 |
| 1 | Woman's age at pregnancy: | | |
| | < 20 years | 1.5 (0.8, 2.7) | 1.3 (0.7, 2.4) |
| | 20–34 years (Ref.) | – | – |
| | ≥ 35 years | 1.8 (1.2, 2.7) | 2.2 (1.4, 3.4) |
| | Parity: | | |
| | No previous live birth | 1.8 (1.1, 2.7) | 4.3 (2.3, 8.0) |
| | 1–2 previous live births (Ref.) | – | – |
| | 3–4 previous live births | 1.2 (0.8, 1.9) | 0.8 (0.5, 1.3) |
| | ≥ 5 previous live births | 1.1 (0.7, 1.8) | 0.7 (0.4, 1.2) |
| | Prior history of pregnancy loss: | | |
| | None (Ref.) | – | – |
| | One or more | 1.3 (1.0, 1.9) | 0.7 (0.5, 1.1) |
| | Ever used family planning: | | |
| | No (Ref.) | – | – |
| | Yes | 0.3 (0.2, 0.5) | 0.2 (0.1, 0.4) |
| | Women's schooling: | | |
| | No schooling | 3.6 (1.5, 8.8) | 1.9 (1.0, 3.9) |
| | Less than secondary | 3.2 (1.3, 7.4) | 1.4 (0.7, 2.9) |
| | Secondary or higher (Ref.) | – | – |
| | Husband's schooling: | | |
| | No schooling | 1.4 (1.0, 2.1) | 2.0 (1.7, 3.4) |
| | Less than secondary | 0.9 (0.6, 1.4) | 0.9 (0.6, 1.6) |
| | Secondary or higher (Ref.) | – | – |
| | Wealth quintile: | | |
| | Highest 20% (Ref.) | – | – |
| | Middle 60% | 0.8 (0.5, 1.3) | 1.5 (0.6, 3.7) |
| | Lowest 20% | 0.7 (0.4, 1.3) | 2.0 (0.9, 4.4) |
| | Residence: | | |
| | Urban (Ref.) | – | – |
| | Rural | 1.2 (0.8, 1.7) | 1.0 (0.7, 1.4) |

Note (1): The odds ratio for each variable adjusted for all the other variables shown in this table (age, parity, history of prior pregnancy loss, ever use of family planning, woman's schooling, husband's schooling, wealth quintile, and urban/rural residence).

*Figures of DHS 2007 have been reproduced from the NIPS report of in-depth analysis of the DHS 2007 data (NIPS 2009) [7].

factors of maternal mortality found in the two surveys. There were no significant changes in the risk of maternal mortality associated with the biological risk factors (woman's age at pregnancy, first birth, prior history of a pregnancy loss, and ever use of family planning). Similarly, the adjusted odds ratios (AORs) were similar for urban/rural residence and wealth quintile (Table 1).

In 2007, women who had less than secondary-level schooling were found to be at a significantly higher risk of maternal mortality than the women who had completed secondary school education or above; this difference disappeared in 2019. On the other hand, in 2007, the

**Table 2. Comparison of the association between maternal mortality and antenatal care and skilled birth attendance in the last pregnancy; Pakistan 2007 and 2019 (see notes 2, 3).**

| Model | Risk Factor | Adjusted Odds Ratio (95% CI) | |
|---|---|---|---|
| | | PDHS 2007* | PMMS 2019 |
| 2 | Antenatal care (last pregnancy): | | |
| | No (Ref.) | – | – |
| | Yes | 0.2 (0.1, 0.4) | 9.3 (4.8, 17.7) |
| 3 | Skilled birth attendance: | | |
| | No (Ref.) | – | – |
| | Yes | 2.2 (1.3, 3.6) | 1.1 (0.7, 1.8) |

Note (2): Odds ratio for each variable adjusted for age, parity, history of prior pregnancy loss, ever use of family planning, woman's schooling, husband's schooling, wealth quintile, and urban/rural residence. Antenatal care is defined as one or more visits to a healthcare provider during pregnancy.

Note (3): Excludes pregnancies not resulting in a live birth.

*Figures of DHS 2007 have been reproduced from the NIPS report of in-depth analysis of the DHS 2007 data (NIPS 2009) [7].

husband's education had no association with maternal mortality but in 2019, it was found that the women whose husbands had no schooling were at a greater risk of maternal mortality than those whose husbands had completed secondary school education or above (Table 1).

Table 2 presents an interesting shift between 2007 and 2019: In 2007, there was no association between antenatal care (ANC) in the last pregnancy and maternal mortality. In 2019, however, women having ANC were found to be at a significantly greater risk of maternal mortality than those not having any ANC. On the contrary, women having skilled birth attendance (SBA) in 2007 were at a greater risk of maternal death but in 2019, there was no association between SBA and maternal mortality.

The impact of distance to hospital and public transportation (nearest city/town from where public transport is available) and maternal mortality (in the rural clusters only) is shown in Table 3. In 2007, women residing in villages that were 40 kilometers or farther from a hospital

**Table 3. Comparison of the association between maternal mortality in rural areas and distance to a hospital providing CEmONC and distance to public transport, Pakistan 2007 and 2019 (see note 4).**

| Model | Risk Factor | Adjusted Odds Ratio (95% CI) | |
|---|---|---|---|
| | | PDHS 2007* | PMMS 2019 |
| 4 | Distance to hospital: | | |
| | < 10 kilometers (Ref.) | – | – |
| | 10–39 kilometers | 1.1 (0.8, 1.7) | 0.6 (0.3, 1.3) |
| | ≥ 40 kilometers | 2.4 (1.5, 4.0) | 0.7 (0.4, 1.5) |
| 5 | Distance to public transport: | | |
| | < 10 kilometers (Ref.) | – | – |
| | 10–39 kilometers | 1.8 (0.9, 3.7) | 1.0 (0.7, 1.6) |
| | ≥ 40 kilometers | 3.6 (1.5, 9.1) | 1.1 (0.6, 2.1) |

Note (4): Both models exclude urban clusters. The odds ratio for each variable adjusted for age, parity, history of prior pregnancy loss, ever use of family planning, woman's schooling, husband's schooling, wealth quintile, and urban/rural residence.

*Figures of DHS 2007 have been reproduced from the NIPS report of in-depth analysis of the DHS 2007 data (NIPS 2009) [7, 18].

or public transport service were at a significantly greater risk of maternal mortality than those residing within 40 kilometers of these services. However, in 2019, there was no association between distance to hospital and/or public transport facilities and maternal mortality.

Finally, in 2007, it was also found that the rural women residing in villages covered by mobile phone service were at a significantly lower risk of maternal mortality compared to those residing outside a mobile phone service range (AOR 0.6; 95% CI 0.4–0.9). This variable was not included in the 2019 analysis, because by that year mobile phones had become universal (over 95% of the households in the sample reported that they owned at least one mobile phone).

## Discussion

Pakistan's modest decline in MMR between 2007 and 2019 can be explained by advancements in communication, transportation, and infrastructure, and some noteworthy changes in healthcare delivery systems that have improved the accessibility and caliber of MNCH care, especially in rural areas.

According to the Demographic Health Surveys 2018, there were evident increases in the uptakes of SBA and ANC (defined as the percentage of pregnant women who had at least four ANC visits). There was an increase in SBA from 34% in 2007 to 66% in 2018 while the increase in ANC was from 28% in 2007 to 51% in 2018 [23]. The two indicators reflect an increase in access and utilization of MNCH care, resulting in a decline in the MMR. Indeed, the decrease in MMR between 2007 and 2019 does not seem to be commensurate with the increases in SBA and ANC, which can only be due to a lack of high-quality MNCH care being provided to women.

Although the health status of Pakistan's population has improved over the years, the pace and the magnitude of the development have been slow. Pakistan's MNCH indicators are worse than the majority of low-income nations including the countries with smaller gross national products per capita [9, 20, 23, 24]. The MMR in Pakistan ranks on the higher side among countries in South Asia [25, 26] while the neonatal mortality rate (NMR) has stalled since 1990 and remains significantly higher than those of other South Asian nations [17]. Many experts have identified inadequate investment in MNCH and economic crises as the primary causes of the current state of affairs [27, 28].

Pakistan is unlikely to meet the SDGs target of reducing its maternal mortality rate to 140 per 100,000 live births by 2030, let alone reach the ultimate goal of 70 [5]. In contrast, neighboring countries such as India have achieved a greater decline in MMR, reaching 103/100,000 [29–31]. Pakistan might benefit from studying and analyzing the policies and programs of other developing nations [32, 33].

A comparison of the causes of maternal deaths confirms that their pattern has remained about the same. There have been proportionate increases in deaths due to PPH and complications of abortion but the MMR attributable to PPH has declined and that due to abortion complications has slightly increased. On the other hand, the percentage of maternal deaths and the MMR due to pregnancy-related hypertension have both increased. These three causes account for 80% of all maternal deaths in Pakistan, which makes it evident that comprehensive EmONC services in both public and private health institutions need to be provided with better quality, accessibility, and affordability [34]. The most common cause of maternal deaths in Latin America was also found to be HDP (25.7%), which has increased in Pakistan over the last two decades.

Maternal age and parity are the two main biological risk factors of maternal mortality [35]. Women having their first birth and those over 34 years are at a higher risk of maternal

mortality. Regardless of the level of development index of a country, pregnancies among women who are comparatively elderly are always fraught with danger. A study in the USA found an increased risk of maternal mortality in women of the older age group, registering a four times increase in the MMR among women aged 40 years and above [36]. Several other studies have found that maternal age of < 20 years and older than 35 years were associated with a higher risk of maternal death [37, 38]. Although the biological risk of complications during the first birth is higher, it can be reduced with proper ANC and closely supervised delivery [40].

Previous pregnancy loss history has been considered a risk factor for both maternal mortality and poor pregnancy outcomes in the future gestation. According to a Norwegian study, women who have previously had miscarriage are more likely to experience it again [39]. In the present study, the impact of prior pregnancy loss disappeared after adjusting for the other biological and socioeconomic risk factors, suggesting that for women with such a history, access to high-quality MNCH care may be able to mitigate the biological risk of maternal problems.

It is important to note that in both PDHS 2007 and PMMS 2019, the risk of maternal death was considerably lower for women who had ever used family planning. This is a significant finding because it shows that women who had previously tried to control their fertility were less likely to die of maternal causes. However, this finding is hardly surprising and several studies show that family planning leads to better reproductive health for women [40–44].

This analysis confirms that, after controlling for the other biological and socio-demographic factors, a poor obstetric history, low socioeconomic status, and residence in rural areas are not significant predictors of maternal mortality. Another interesting finding was that in 2007, women's lack of schooling was an important predictor of maternal mortality while in 2019 it was not the case. Even though that earlier research studies found that a higher level of female education, resulted in female autonomy and a greater likelihood of use of modern health services, reflecting a reduction in the risk of maternal complications [45]. Conversely, in 2007 the husband's level of education had no bearing on the likelihood of maternal death; nevertheless, by 2019, the husband's lack of education was a major predictor. A possible explanation of this seemingly paradoxical finding is that the women's ability to access EmONC services and possibly a reduction in the first and second delays in accessing EmONC override the impact of female education.

Analysis of the data in the present study has found no association between maternal mortality and grand multiparity ($\geq$ 5 previous births) neither in 2007 nor in 2019, although other studies have found such an association showing an increasing risk with the increase in the number of previous births [46]. One possible explanation of this apparent anomaly may be that some of the other risks of maternal mortality for other reasons override the risk of grand multiparity, although there is no direct evidence to support this hypothesis.

Another interesting finding was that in 2007 but not in 2019, the risk of maternal death was significantly impacted by the distance to a hospital and/or public transit facility, which indicates improved access to MNCH services in rural areas. The outsourcing of the government's basic health units and rural health centers to outside organizations is a significant step in this regard since it has significantly improved the quality of care in these facilities.

Developing comprehensive interventions to minimize preventable maternal can be made easier with an awareness of the underlying causes and risk factors [47]. The MNCH interventions in Pakistan must include an effective family planning service delivery, emphasis on skilled birth attendance, a focus on improving the quality of care at all stages of the continuum of care, and on the provision of quality MNCH care to all women regardless of their residence, education, or socioeconomic status. In this regard, the current initiatives of the Lady Health Workers, Community Midwives, outsourcing of public sector primary health care facilities,

social health insurance, universal health coverage, and establishment of provincial quality of care commissions are the steps in the right direction. There is a need to strengthen these initiatives and ensure that quality MNCH services are available across the country in urban and rural areas in both the public and private sectors. Family planning plays a central role in reducing maternal mortality; unfortunately, in Pakistan, family planning is not a priority agenda despite its very rapid population growth. A restructuring of the family planning program to reach the target of a contraceptive prevalence rate of 55% would help a faster decline in MMR.

In addition to the above, the government needs to strengthen the National Committee for Maternal and Neonatal Health (NCMNH), which offers technical support in policymaking and raising awareness about maternal health [17].

The current initiatives of the Community Midwives, Lady Health Workers, social health insurance, universal health coverage, outsourcing of public sector primary health care facilities, and the creation of provincial quality of care commissioners are all positive steps in this direction. These programs must be strengthened to guarantee that the public and private sectors in both urban and rural areas of the country offer high-quality MNCH services. Family planning plays a central role in reducing maternal mortality; unfortunately, in Pakistan, family planning is not a priority agenda despite its very rapid population growth. A restructuring of the family planning program to reach the target of a contraceptive prevalence rate of 55% would help a faster decline in MMR.

## Conclusions

Pakistan has experienced a modest decline in MMR between 2007 and 2019, which can be attributed to an increase in SBA and ANC due to improved access to primary health care and Emergency Obstetrics and Neonatal Care (EmONC) services, particularly in the rural areas. Pakistan has the running programs and available resources to enhance the impact of these programs by further strengthening the healthcare delivery system and ensuring a reasonably large increase in family planning. The further steps are to enhance the capabilities of female birth attendants and the contributions of nonprofit organizations.

## Limitation of the study

The present analysis was done by making a comparison of the secondary data of PDHS 2007 and PMMS 2019, which was limited to the questionnaires and sample sizes based on the previously determined framework. Although there were numerous factors affecting the outcome of a pregnancy, this study focused only on a few key factors for comparison which were common in both surveys.

Despite the detailed analysis of the dataset, it was not possible to capture all the variables that may influence MMR, including maternal nutritional status, the time of referral, the provision of services, and access to the EmONC facilities. Also, the capacity to determine a precise cause of death was constrained because of the secondary analysis of the data.

## Supporting information

**S1 File. SPSS data analysis files.**
(XLSX)

## Acknowledgments

We extend our thanks to the National Institute of Population Studies (NIPS) and the DHS Program for collecting and providing the data that made this study possible.

## Author Contributions

**Conceptualization:** Farid Midhet, Samina Naeem Khalid, Shehla Baqai, Shahzad Ali Khan.

**Data curation:** Farid Midhet.

**Formal analysis:** Farid Midhet.

**Methodology:** Farid Midhet.

**Resources:** Samina Naeem Khalid.

**Software:** Farid Midhet.

**Supervision:** Shehla Baqai, Shahzad Ali Khan.

**Visualization:** Samina Naeem Khalid, Shehla Baqai.

**Writing – original draft:** Farid Midhet, Samina Naeem Khalid, Shehla Baqai.

**Writing – review & editing:** Samina Naeem Khalid, Shahzad Ali Khan.

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
