## [Decision Letter · Decision Letter 0]

24 Sep 2024

Trends in the Levels, Causes, and Risk Factors of Maternal Mortality in Pakistan: A Comparative Analysis of National Surveys of 2007 and 2019.

PONE-D-24-01507

Dear Dr. Khalid,

We’re pleased to inform you that your manuscript has been judged scientifically suitable for publication and will be formally accepted for publication once it meets all outstanding technical requirements.

Kind regards,

Fernanda Penido Matozinhos, Ph.D

Academic Editor

PLOS ONE

Additional Editor Comments (optional):

Dear Author,

After careful consideration, I feel the manuscript explores a very important topic and it fully meet PLOS ONE’s publication criteria as it currently stands.

Kind regards,

Reviewers' comments:

Reviewer's Responses to Questions

**Comments to the Author**

1. Is the manuscript technically sound, and do the data support the conclusions?

Reviewer #1: Yes

2. Has the statistical analysis been performed appropriately and rigorously? 

Reviewer #1: Yes

3. Have the authors made all data underlying the findings in their manuscript fully available?

Reviewer #1: Yes

4. Is the manuscript presented in an intelligible fashion and written in standard English?

Reviewer #1: Yes

5. Review Comments to the Author

Reviewer #1: This study has looked in depth at the into the trends and risk factors associated with maternal mortality in Pakistan from 2007 to 2019, against the backdrop of global efforts to achieve Sustainable Development Goal targets for maternal health. Employing secondary data analysis, the study compares results from the Pakistan Maternal Mortality Survey 2019 with the Pakistan Demographic and Health Survey 2007, employing a nested case-control design. Logistic regression, adjusting for crucial factors, is utilized to estimate odds ratios for major risk factors of maternal mortality. The findings reveal a 33% reduction in Pakistan's Maternal Mortality Ratio (MMR) to 186 per 100,000 live births in 2019, with notable rural-urban disparities. Leading causes of maternal mortality include postpartum hemorrhage, hypertensive disease of pregnancy, postpartum infection, and post-abortion complications. Advanced maternal age (>35 years) and primiparity emerge as risk factors, while family planning appears protective. Access to quality emergency obstetric care remains pivotal, evidenced by deaths occurring despite antenatal care. The study underscores the persistent influence of limited family planning on maternal mortality and emphasizes the need for targeted interventions to address access barriers and ensure comprehensive maternal health care in Pakistan.

6. PLOS authors have the option to publish the peer review history of their article (what does this mean?). If published, this will include your full peer review and any attached files.

Reviewer #1: **Yes: **Mariyam Sarfraz

---

## [Editor Report · Acceptance letter]

30 Sep 2024

PONE-D-24-01507 

PLOS ONE

Dear Dr. Khalid, 

I'm pleased to inform you that your manuscript has been deemed suitable for publication in PLOS ONE. Congratulations! Your manuscript is now being handed over to our production team.

Kind regards, 

on behalf of

Dr. Fernanda Penido Matozinhos 

Academic Editor

PLOS ONE